# Assessment of Recovered Struvite as a Safe and Sustainable Phosphorous Fertilizer

Carolina Mancho * , Sergio Diez-Pascual, Juan Alonso, Mar Gil-Díaz and M. Carmen Lobo

Instituto Madrileño de Investigación y Desarrollo Rural, Agrario y Alimentario (IMIDRA), Finca "El Encín", A-2, km 38,5. Alcalá de Henares, 28805 Madrid, Spain
* Correspondence: carolina.mancho@madrid.org

**Abstract:** Phosphorus (P) is an essential nutrient for crops. Modern agriculture is dependent on P derived from phosphate rock, which is a non-renewable resource that is becoming increasingly scarce and expensive. Therefore, it is necessary to identify alternative sources of P and develop novel technologies for its recovery. Such technologies are increasingly focused on the recovery of struvite ($MgNH_4PO_4 \cdot 6H_2O$) (STR) from urban wastewater. A reduction of P in effluents decreases the risk of eutrophication while allowing this mineral to be recovered and reused. Here we applied STR recovered from urban wastewater to three different soils and examined its slow-release properties. We performed a soil column leaching study and compared the results of soil treated with STR with those of soil amended with conventional phosphorous fertilizers, namely NPK, ammonium phosphate (MAP), and superphosphate (SSP). Phosphate leaching capacity followed the order MAP ~ NPK > SSP > STR in the three soils and was consistent with its rate of water solubility. Analysis of the soils after the leaching process showed differences in available P, the highest content being found in soils treated with STR. The results were consistent with those obtained in the leaching assay. STR behaves as a slow-release fertilizer and reduces the risk of eutrophication compared to the conventional P fertilizers tested.

**Keywords:** agricultural soil; P; fertilization; leaching assay; eutrophication

---

## 1. Introduction

Phosphorus (P), like nitrogen (N) and potassium (K), is an essential plant nutrient. In this regard, P is a structural component of phospholipids, nucleic acids, nucleotides, coenzymes, and phosphoproteins. It participates in protein synthesis and facilitates cell division and the development of new tissues, thereby contributing to the transport, storage, and transfer of energy [1]. Fertilizer demand is expected to rise in the long term due to the increase in the global population, which will reach 9.8 billion people by 2050, according to the United Nations forecast [2]. World crop production could increase by 50% to 69% during the 2010–2050 period [3]. Other authors even predict an increase between 100 and 110% in global crop demand in the same period [4]. The world's main source of P (phosphate rock) is a non-renewable resource, and high-quality reserves are increasingly scarce and expensive [5]. The consumption of P is rising by 1.5% per year, and reserves of this mineral are expected to diminish within the next 90 to 300 years if this rate is maintained [6]. This panorama raises issues related to the acquisition of P fertilizers and threats to future food security, especially in countries with no reserves of this mineral [7,8]. In this regard, Spain, like other European countries, has no reserves of phosphate rock and is dependent on the imported mineral P fertilizers to sustain agriculture and ensure food security.

The list of Critical Raw Materials for the EU includes both P and phosphates, which are considered essential for the production of a wide range of products and services for daily use. This list, which includes 27 raw materials, incentivizes Europe to produce these materials through enhancing recovery, reuse, and recycling activities [9]. According to this



document, China is the main provider of P and phosphate rock (55% and 44%, respectively). However, Kazakhstan and Morocco are the main suppliers of P (77%) and phosphate rock (27%) to the EU, respectively. P extraction has significantly decreased the natural rock reserves of this mineral [10,11], and global reserves may be depleted in 50–100 years at the current rate of extraction [12,13]. Most global demand for P is for fertilizer and food production [11,14]. In agriculture, other sources of P, such as bone meal, crop residues, and manure, can be used. However, these are insufficient in terms of complete resource circulation and sustainable retention of nutrients. In this context, the development of new technologies for the sustainable recovery of P from other organic products is, therefore, required [15].

Several technologies could cover the demand for phosphate fertilizers for global food production. Municipal wastewater is a promising source of P via its reuse and could serve to replace P derived from phosphate rock. Sewage sludge has traditionally served as a source of P for agricultural land. However, in several European countries, its use for this purpose has either been restricted by legislation or stopped because of environmental risks posed by pollutants such as heavy metals and organic compounds. Therefore, in recent years, several technologies have been developed to recover P from wastewater [16]. These efforts are increasingly focused on the recovery of struvite ($MgNH_4PO_4\cdot6H_2O$) (STR) from urban wastewater. Several authors have reported that the municipal wastewater of countries in Central Europe contains a P load that could theoretically replace 40 to 50% of the mineral P fertilizer applied each year in agriculture [17,18]. Access to wastewater is already guaranteed, and technologies to extract P from this waste reduce the risk of eutrophication while at the same time allowing the recovery and reuse of this mineral [19,20]. Thus, Directive 91/271/EEC [21], concerning urban wastewater treatment, and the EU Water Framework Directive [22] both require that potential pollutants be removed from wastewater before its discharge to surface water, thereby limiting N and P levels and reducing the risk of eutrophication of sensitive water. In addition, methods to recover P during wastewater treatment, such as the precipitation of STR under controlled conditions, improve the maintenance of wastewater treatment plants by preventing unwanted crystal formation, which can otherwise clog and damage these installations [23]. Thus, the inclusion of such a recovery process as part of a wastewater management system would allow savings in maintenance costs associated with uncontrolled STR precipitation, and sludge reduction, as well as environmental benefits linked to a decrease in P discharges and the recovery and recycling of this mineral [24,25]. These recovery techniques are integrated into the framework of the New Circular Economy Action Plan of the EU, which advocates for the development of an Integrated Nutrient Management Plan, with a focus on ensuring a more sustainable application of nutrients and stimulating markets for recovered nutrients. Currently, European legislation (EU 2021/2086 of the European Commission) includes the salts of precipitated phosphate and its derivatives as a category of component materials authorized in EU fertilizers, being accepted sewage sludge from municipal wastewater treatment plants as raw materials for its production.

The properties of STR make it suitable as a fertilizer [26–30]. Its most advantageous characteristic as a fertilizer is its slow rate of nutrient release. This property allows for a direct and higher application dose of STR, exceeding those of conventional fertilizers without causing adverse effects on plant health [31–33]. STR is applied mainly as a P fertilizer since its low $N/P_2O_5$ ratio makes N insufficient for optimal plant growth. In agricultural practice, the required amount of N is far higher than that of P, so it is often convenient to supplement STR with other N sources [27,34]. In this regard, most studies indicate that STR has an agronomic efficiency similar to that of fertilizers derived from phosphoric rock and processed P fertilizers [35]. Several studies addressing STR recovered from diverse organic wastes have focused on plant production and have reported positive results in a variety of crops, including lettuce [36–38], maize [26,34], ryegrass [39], lupin [40], Chinese cabbage [41] and wheat [42,43].

Soil properties can condition the behavior of fertilizer. However, to the best of our knowledge, there have been no studies on the leaching of salts caused by the application of phosphorous fertilizers in different types of soils. To address this question, here, we used a soil column leaching assay to compare the slow-release properties of STR in three distinct agricultural soils with the performance of other traditional P fertilizers.

## 2. Materials and Methods

### 2.1. Column Leaching Assay

To assess the potential leaching of soluble phosphate salts after applying P fertilizers to the soil, a column leaching assay was performed using three agricultural soils with different pHs (acidic, neutral, and alkaline) taken from Central Spain. The soils used are representative of the Mediterranean climate. Their physicochemical characteristics are shown in Table 1. Granular struvite (STR) and three commercial P fertilizers: granular NPK, powdered monoammonium phosphate (MAP), and granular single superphosphate (SSP), were tested. The physicochemical characteristics of the fertilizers and their heavy metals content are provided in Table 2. In addition, an untreated control soil was used. The STR was from the Canal de Isabel II South Wastewater Treatment Plant in Madrid (Spain). Glass columns, 30 cm in height and 4.5 cm in internal diameter were prepared per treatment and soil. They were closed at the bottom with a filtering membrane and a layer of fine gravel to prevent soil compaction. Each column was filled with 400 g of dried and sieved soil. Previously, the soil corresponding to the superficial layer (10 cm) was treated with fertilizers. The dose used corresponded to an equivalent concentration of 100 kg P ha$^{-1}$ (229 kg P$_2$O$_5$ ha$^{-1}$), resulting in an application of 30.53 mg of P$_2$O$_5$ per column. The soils were saturated with distilled water, and 100 mL of water was subsequently applied every day to force gravity leaching. Leachates were collected every 24 h. The total volume of water added corresponded to two years of rainfall in central Spain (approx. 1000 mL in each column). Immediately after each leaching event, the pH and electrical conductivity (EC) were determined. Leachates were filtered, and PO$_4^{3-}$, Cl$^-$, NO$_3^{2-}$, and SO$_4^{2-}$ were then determined using ionic chromatography (ICS-1100, Dionex, Camberley, UK). Ca$^{2+}$, Mg$^{2+}$, Na$^+$, and K$^+$ were quantified using flame atomic absorption spectrometry (AA240FS, Varian, Mulgrave, Australia).

**Table 1.** Physicochemical characteristics of soils.

| Soil | pH | EC (dS/m) | N (%) | OM (%) | P | Ca | Mg | Na | K | Pb (mg/kg) | Cd | Cu | Ni | Zn | Cr | Sand | Silt (%) | Clay |
|------|-----|-----------|-------|--------|-----|------|-----|----|-----|-----|-----|-----|-----|----|----|------|------|------|
| Acidic | 5.15 | 0.67 | 0.06 | 0.83 | 42 | 751 | 103 | 21 | 144 | <DL | <DL | <DL | <DL | 8 | 17 | 24 | 64 | 12 |
| Neutral | 7.66 | 0.22 | 0.15 | 3.23 | 83 | 1943 | 121 | 23 | 210 | <DL | <DL | <DL | <DL | 10 | 22 | 47 | 41 | 12 |
| Alkaline | 8.57 | 0.22 | 0.11 | 1.49 | 29 | 2877 | 310 | 17 | 379 | 19 | <DL | 16 | 13 | 52 | 57 | 58 | 4 | 38 |

Note: EC: Electrical conductivity; OM: Organic matter; DL: Detection level.

Finally, *P losses* in the leaching process for each fertilizer were calculated as mg of PO$_4^{3-}$ in the total volume leached from each column and as a percentage of the dose applied.

$$Fertilizer\ P\ loss\ (\%) = \frac{P\ leached_{fert} - P\ leached_{ctr}}{applied\ P} \times 100$$

where *P leached(fert)* is the total loss of PO$_4^{3-}$ in the entire leaching process of the fertilized soils, and *P leached(ctr)* is the total loss of PO$_4^{3-}$ from the control soil after the same process.

**Table 2.** Main physicochemical characteristics and heavy metals content of the fertilizers.

|  | STR | NPK | MAP | SSP |
|---|---|---|---|---|
| Water soluble $P_2O_5$ (%) | 1.3 | 14.1 | 46.5 | 17.5 |
| Citrate-soluble $P_2O_5$ (%) | 22.3 | 15.2 | 61.6 | 18.0 |
| Soluble in mineral acids $P_2O_5$ (%) | 28.8 | 15.1 | 61.5 | 19.2 |
| Total N (%) | 5.7 | 14.4 | 12.2 | 0.5 |
| Ammonium N (%) | 5.5 | 14.1 | 11.9 | 0.3 |
| Nitrate N (%) | 0.1 | 0.2 | 0.2 | 0.1 |
| Ureic N (%) | <1.0 | <1.0 | <1.0 | <1.0 |
| K water soluble (in K2O) % | <1.0 | 15.3 | <1 | <1 |
| Cd (mg/kg DM) | <0.5 | <0.5 | <0.5 | 18.5 |
| Cu (mg/kg DM) | <20.0 | <20.0 | <20.0 | 21.5 |
| Cr (mg/kg DM) | <10.0 | <10.0 | <10.0 | 38.0 |
| Hg (mg/kg DM) | <0.4 | <0.4 | <0.4 | <0.4 |
| Ni (mg/kg DM) | <5.0 | <5.0 | <5.0 | 31.6 |
| Pb(mg/kg DM) | <5.0 | <5.0 | <5.0 | <5.0 |
| Zn (mg/kg DM) | <25.0 | <25.0 | <25.0 | 286 |
| As (mg/kg DM) | <2.0 | <2.0 | <2.0 | 5 |
| B (mg/kg DM) | <4.0 | <4.0 | <4.0 | 19.2 |
| Mo (mg/kg DM) | <0.5 | <0.5 | <0.5 | 17.9 |
| Mn (mg/kg DM) | 36.2 | 142.0 | <10.0 | 20.5 |

*2.2. Soil Analysis after Leaching Process*

After the leaching, the soil from each column was collected, air-dried, and sieved (<2 mm) prior to analysis. Four samples per treatment were analyzed. The physico-chemical properties of the soil were determined following the official methodology in Spain (MAPA, 1994). Organic matter was determined using the Walkley–Black method. pH and EC were measured in a 1:2.5 soil:water ratio, the total N content was quantified using the Kjeldahl method, and available nutrients (Ca, K, Mg, Na) were extracted with 0.1 N ammonium acetate and quantified using FAAS (AA240FS, Varian). Heavy metal concentrations in the soil samples were determined after acid digestion in a microwave reaction system (Multiwave Go, Anton Paar GmbH). In the digestion extract, the concentrations of Cd, Cr, Cu, Ni, Pb, and Zn were quantified with flame atomic absorption spectrometry.

*2.3. Statistical Analysis*

The data were analyzed using version 24.0 of the SPSS program for Windows. Differences between treatments were evaluated with one-way analysis of variance (ANOVA) at a significance level of $p < 0.05$. The mean values of the replicates were compared using Duncan's test.

## 3. Results

*3.1. pH and EC of Leachate*

In the alkaline soil, the pH of the first leachate in the treatments ranged between 8.00 and 8.18 (Figure 1A) and tended to increase as leaching progressed. However, the pH of the last leachate did not differ significantly from the first one in any of the other treatments. The first leachates had the highest EC values in all treatments, being significantly higher in the case of NPK in relation to the control and the rest of the fertilizers. Throughout the process, the leachates from the NPK and SSP treatments showed the highest EC values. However, at the end of the process, no significant differences were found in the EC of the leachates from the different treatments.

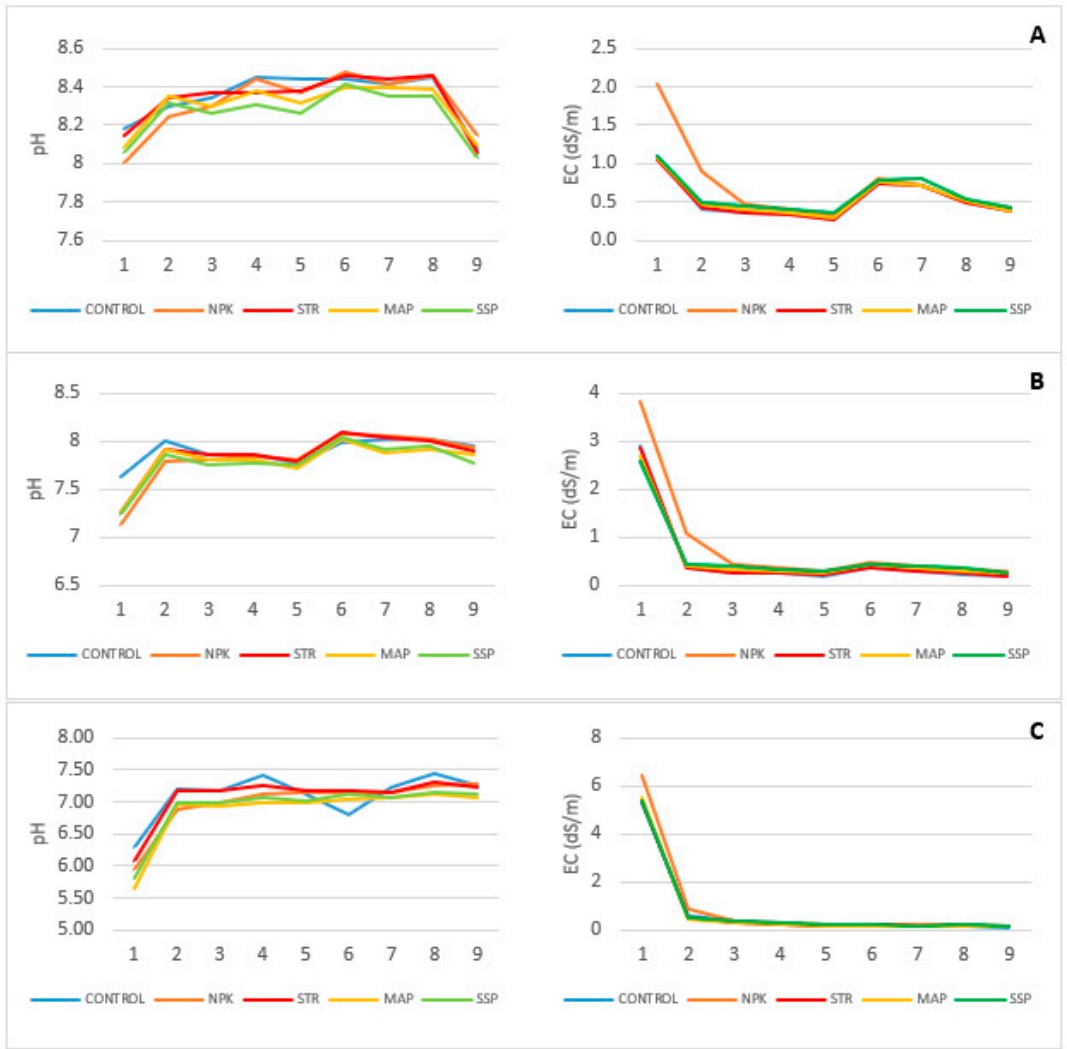

**Figure 1.** pH and electrical conductivity (dS/m) of leachates in alkaline (**A**), neutral (**B**), and acidic (**C**) soil.

In the neutral soil, the pH of the first leachate in the treatments ranged between 7.12 and 7.35 and increased as leaching progressed. The pH of the final leachates ranged between 7.77 and 7.95 and was statistically higher than the initial pH in all treatments (Figure 1B). In this soil, the first leachate also had the highest EC values of all the treatments, being significantly higher in the case of NPK. Over the course of the leaching process, no significant differences were observed among treatments.

In the acidic soil, the pH of the first leachate in the treatments ranged between 5.65 and 6.28 and increased progressively over time in all treatments, with the pH of the final leachate being significantly higher than the initial one and ranging between 7.06 and 7.28 (Figure 1C). The leachates obtained from the control, STR- and NPK-treated soils at the end of the process had the highest pH values. In the acidic soil, the first leachates had the highest EC values in all cases, being significantly higher in the case of NPK in relation to the rest of the fertilizers. From leachate 4 onwards, leachates from the NPK and SSP treatments had the highest EC values and remained statistically higher than the other treatments until the end of the leaching process.

*3.2. Phosphorus in Leachate*

In the alkaline soil, P fertilizers showed a similar behavior (Figure 2A). Soil fertilized with NPK leached the highest amount of P, 1.32 mg $PO_4^{3-}$, while the control, STR-, MAP-

and SSP-treated soils leached 1.13, 1.16, 1.13, and 1.17 mg $PO_4^{3-}$, respectively. These observations imply a loss of 0.61, 0.10, 0.01, and 0.11% of the P applied with NPK, STR, MAP, and SSP, respectively (Table 3). The differences between the treatments were not statistically significant.

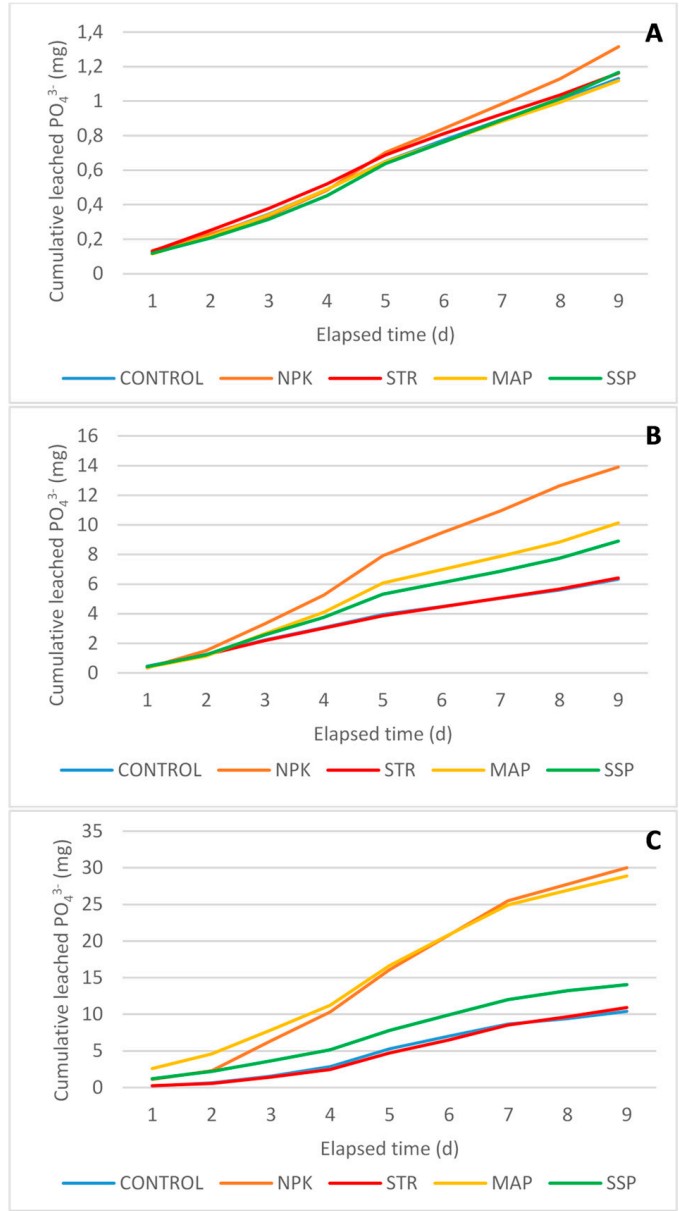

**Figure 2.** Amount of P salts leached from alkaline (**A**), neutral (**B**), and acidic (**C**) soil columns as a function of time in struvite and P fertilizers.

**Table 3.** P losses (%) in the different soils for each fertilizer.

|  | Alkaline Soil | Neutral Soil | Acidic Soil |
| --- | --- | --- | --- |
| NPK | 0.61 | 24.80 | 64.24 |
| STR | 0.10 | 0.24 | 1.72 |
| MAP | 0.10 | 12.42 | 60.65 |
| SSP | 0.11 | 8.46 | 11.93 |

In neutral pH soil, NPK caused the highest amount of P leaching (13.9 mg), with statistically significant differences with respect to the rest of the fertilizers, followed by

MAP, SSP, STR, and the control (10.12, 8.91, 6.41, and 6.34 mg, respectively). Soil treated with MAP also leached significantly more P than the control and STR-treated soil. Leaching resulted in a P loss of 24.80, 0.24, 12.42, and 8.46% for the NPK, STR, MAP, and SSP treatments, respectively (Table 3). The highest P leaching occurred towards the middle of the process, decreasing thereafter in all treatments (Figure 2B).

In the acidic soil, NPK and MAP produced the highest P leaching (30 and 28.91 mg, respectively), the differences being statistically significant, followed by SSP, STR, and the control (14.05, 10.93, and 10.41, respectively) (Table 3). These observations represent a loss of 64.24, 1.72, 60.65, and 11.93% of the P applied with NPK, STR, MAP, and SSP, respectively.

### 3.3. Nitrate, Chloride, and Sulfate in Leachates

In alkaline, neutral, and acidic soil, nitrate leached mainly during the first leachate, with remaining leaching rates low in the subsequent leachates (Figure S1, in Supplementary Material). In the alkaline soil, nitrate leaching was similar for all the treatments assayed. However, neutral and acidic soils treated with NPK and MAP showed significantly higher nitrate leaching than the rest of the treatments (Table 4). The highest nitrate leaching was observed in the acidic soil (337.83–358.52 mg), followed by the alkaline (33.78–38.32 mg) and neutral soils (88.02–112.87 mg).

Chloride behaved similarly to nitrate, with the highest amount of these salts lost in the first leachate in all soils (Figure S2, in Supplementary Material). NPK was the fertilizer that caused the highest amount of chloride leaching in the acidic (43.58 mg), alkaline (26.56 mg), and neutral (17.37 mg) soils, the differences being statistically significant in all cases. NPK was followed by SSP, which showed significantly higher chloride leaching than the control, STR, and MAP treatments in the acidic (15.41 mg) and neutral (5.75 mg) soils (Table 4).

Similarly, the highest sulfate leaching occurred in the NPK treatment in the acidic (126.96 mg), alkaline (29.31 mg), and neutral (39.94 mg) soils, with significant differences compared to the other treatments (Figure S3, in Supplementary Material). As in the case of chloride, SSP showed significantly higher sulfate leaching than the control, STR, and MAP treatments in the acidic (79.08 mg), alkaline (29.31 mg), and neutral (26.21 mg) soils (Table 4).

### 3.4. Calcium, Magnesium, Sodium, Potassium, and Heavy Metals in Leachates

The NPK treatment showed a significant concentration of K in the leachates in all soil types in comparison to the control soil due to fertilizer composition. Despite the concentration of Mg in STR, an increase in this element in the leachates was not observed (Table 4).

Moreover, the presence of heavy metals was not observed in the leachates from the different treatments despite the content of Cd, Cu, and Cr present in SSP (Table 2).

### 3.5. Soil Analysis after Leaching

After the leaching process in alkaline soil treated with SSP, a significant reduction in pH and an increase in EC were observed with respect to the control. The EC also increased in the NPK-treated soil. All fertilizer treatments significantly increased soil P content, the higher values corresponding to the STR and MAP treatments (Figure 3). No significant differences were observed in relation to other nutrients, such as Ca, Mg, Na, and K, or heavy metals (Table 5).

**Table 4.** Chemical composition of cumulative leaching. Values followed by different letters indicate significant differences (*p* < 0.05, Duncan's test).

| | | $PO_4^{3-}$ (mg) | $NO_3^-$ (mg) | $Cl^-$ (mg) | $SO_4^{2-}$ (mg) | $K^+$ (mg) | $Na^+$ (mg) | $Ca^{2+}$ (mg) | $Mg^{2+}$ (mg) |
|---|---|---|---|---|---|---|---|---|---|
| Alkaline soil | Control | 1.13 ± 0.01 | 34.08 ± 0.79 | 6.77 ± 0.01 a | 5.13 ± 0.07 a | 25.40 ± 1.05 a | 4.62 ± 0.47 | 52.82 ± 0.95 a | 11.93 ± 0.25 a |
| | STR | 1.16 ± 0.02 | 34.75 ± 0.97 | 6.99 ± 0.09 a | 5.14 ± 0.01 a | 25.37 ± 0.50 a | 4.38 ± 0.03 | 54.61 ± 0.54 ab | 12.29 ± 0.17 a |
| | NPK | 1.32 ± 0.23 | 33.78 ± 0.24 | 26.56 ± 3.42 b | 63.4 ± 16.52 c | 31.15 ± 2.16 b | 4.83 ± 0.05 | 86.00 ± 1.13 d | 16.67 ± 0.45 b |
| | MAP | 1.12 ± 0.01 | 38.02 ± 0.47 | 6.77 ± 0.03 a | 5.03 ± 0.17 a | 26.88 ± 1.16 a | 4.61 ± 0.19 | 57.40 ± 1.71 bc | 12.40 ± 0.17 a |
| | SSP | 1.17 ± 0.10 | 34.99 ± 0.23 | 7.93 ± 0.10 a | 29.31 ± 5.15 b | 25.43 ± 0.95 a | 6.87 ± 2.44 | 59.30 ± 0.74 c | 17.44 ± 0.49 b |
| Neutral soil | Control | 6.34 ± 0.06 a | 88.02 ± 1.73 a | 4.43 ± 0.33 a | 4.33 ± 0.02 a | 11.01 ± 1.07 a | 4.44 ± 0.20 a | 67.61 ± 5.13 a | 6.71 ± 0.64 a |
| | STR | 6.41 ± 0.01 a | 100.38 ± 1.68 b | 4.88 ± 0.01 a | 4.32 ± 0.03 a | 17.53 ± 0.35 a | 4.54 ± 0.03 a | 68.46 ± 0.03 a | 7.07 ± 0.05 a |
| | NPK | 13.90 ± 2.16 c | 109.73 ± 3.34 c | 17.37 ± 0.18 c | 39.94 ± 0.13 c | 10.26 ± 0.55 b | 5.47 ± 0.06 c | 102.48 ± 1.37 b | 10.44 ± 0.17 b |
| | MAP | 10.12 ± 0.68 ab | 112.87 ± 1.19 c | 4.82 ± 0.01 a | 4.45 ± 0.26 a | 11.58 ± 0.61 a | 4.94 ± 0.04 b | 71.33 ± 2.69 a | 7.27 ± 0.24 a |
| | SSP | 8.91 ± 0.07 b | 94.45 ± 1.04 ab | 5.75 ± 0.24 b | 26.21 ± 3.86 b | 11.89 ± 0.49 a | 4.94 ± 0.22 b | 70.32 ± 1.00 a | 7.34 ± 0.07 a |
| Acidic soil | Control | 10.41 ± 2.39 a | 340.65 ± 6.56 a | 13.69 ± 0.07 a | 59.62 ± 6.22 a | 18.26 ± 0.08 a | 6.84 ± 0.18 | 85.61 ± 2.91 | 14.34 ± 0.09 |
| | STR | 10.93 ± 0.85 a | 342.51 ± 3.41 a | 14.14 ± 0.49 ab | 60.76 ± 0.78 a | 19.07 ± 0.02 ab | 7.80 ± 0.90 | 82.14 ± 10.48 | 14.38 ± 0.47 |
| | NPK | 30.00 ± 3.17 b | 357.40 ± 6.43 b | 43.58 ± 1.15 c | 126.96 ± 1.11 c | 22.50 ± 1.72 c | 7.28 ± 0.32 | 102.76 ± 4.83 | 20.85 ± 4.71 |
| | MAP | 28.91 ± 3.67 b | 358.52 ± 5.87 b | 15.47 ± 0.04 b | 60.73 ± 3.01 a | 19.92 ± 0.17 ab | 7.36 ± 0.17 | 86.33 ± 0.81 | 13.81 ± 2.08 |
| | SSP | 14.05 ± 2.06 a | 342.14 ± 3.96 a | 15.41 ± 0.37 b | 79.08 ± 2.17 b | 21.06 ± 1.09 bc | 7.62 ± 0.01 | 85.53 ± 1.31 | 14.09 ± 1.25 |

**Table 5.** Physicochemical characteristics of soils after leaching assay. Values followed by different letters indicate significant differences (*p* < 0.05, Duncan's test). DL: detection level.

| | | pH | E.C. | N | M.O. | P | Ca | Mg | Na | K | Pb | Cd | Cu | Ni | Zn | Cr |
|---|---|---|---|---|---|---|---|---|---|---|---|---|---|---|---|---|
| | | | (dS/m) | (%) | (%) | (mg/kg) | | | | | | | | | | |
| Alkaline soil | Control | 8.10 ± 0.04 a | 0.231 ± 0.005 a | 0.131 ± 0.001 | 1.84 ± 0.07 | 37 ± 1 a | 3141 ± 51 | 385 ± 4 | 12 ± 3 | 415 ± 4 | 19 ± 1 | <DL | 16 ± 0 | 13 ± 1 | 50 ± 1 | 45 ± 1 |
| | STR | 8.14 ± 0.05 a | 0.233 ± 0.008 a | 0.132 ± 0.000 | 1.92 ± 0.06 | 66 ± 3 c | 3135 ± 44 | 378 ± 18 | 14 ± 5 | 394 ± 25 | 19 ± 1 | <DL | 17 ± 1 | 13 ± 0 | 52 ± 1 | 45 ± 1 |
| | NPK | 8.09 ± 0.04 a | 0.262 ± 0.006 b | 0.131 ± 0.001 | 1.90 ± 0.08 | 56 ± 2 b | 3055 ± 17 | 349 ± 3 | 18 ± 1 | 423 ± 34 | 18 ± 1 | <DL | 17 ± 1 | 13 ± 1 | 51 ± 1 | 44 ± 5 |
| | MAP | 8.07 ± 0.02 ab | 0.230 ± 0.008 a | 0.133 ± 0.006 | 1.72 ± 0.02 | 60 ± 8 bc | 3257 ± 18 | 390 ± 26 | 16 ± 3 | 386 ± 1 | 18 ± 1 | <DL | 16 ± 0 | 13 ± 0 | 51 ± 0 | 45 ± 1 |
| | SSP | 7.97 ± 0.03 b | 0.301 ± 0.02 c | 0.131 ± 0.004 | 1.78 ± 0.07 | 56 ± 2 b | 2973 ± 45 | 352 ± 18 | 13 ± 6 | 379 ± 23 | 19 ± 1 | <DL | 16 ± 0 | 14 ± 1 | 51 ± 1 | 46 ± 4 |
| Neutral soil | Control | 7.35 ± 0.03 a | 0.230 ± 0.138 | 0.152 ± 0.014 a | 3.38 ± 0.18 | 84 ± 11 a | 1933 ± 63 | 105 ± 1 | 12 ± 0 | 140 ± 4 a | 14 ± 1 | <DL | 11 ± 3 | <DL | 57 ± 4 | <DL |
| | STR | 7.58 ± 0.01 b | 0.132 ± 0.010 | 0.191 ± 0.000 b | 3.52 ± 0.31 | 112 ± 6 c | 1937 ± 86 | 119 ± 7 | 14 ± 2 | 144 ± 3 a | 14 ± 1 | <DL | 9 ± 0 | <DL | 53 ± 3 | <DL |
| | NPK | 7.53 ± 0.03 b | 0.171 ± 0.001 | 0.192 ± 0.000 b | 2.94 ± 0.08 | 87 ± 1 a | 2074 ± 101 | 111 ± 1 | 14 ± 2 | 203 ± 11 b | 13 ± 0 | <DL | 9 ± 0 | <DL | 53 ± 4 | <DL |
| | MAP | 7.59 ± 0.06 b | 0.121 ± 0.007 | 0.174 ± 0.007 ab | 3.26 ± 0.19 | 86 ± 3 a | 2072 ± 10 | 114 ± 4 | 14 ± 4 | 149 ± 1 a | 16 ± 5 | <DL | 10 ± 0 | <DL | 51 ± 6 | <DL |
| | SSP | 7.56 ± 0.04 b | 0.172 ± 0.004 | 0.193 ± 0.014 b | 3.38 ± 0.21 | 102 ± 3 b | 1991 ± 12 | 110 ± 1 | 11 ± 1 | 141 ± 4 a | 16 ± 0 | <DL | 8 ± 0 | <DL | 50 ± 0 | <DL |
| Acidic soil | Control | 5.96 ± 0.08 a | 0.042 ± 0.002 a | 0.056 ± 0.000 a | 0.69 ± 0.03 | 44 ± 1 a | 428 ± 13 | 59 ± 1 a | 9 ± 1 | 97 ± 3 a | 13 ± 0 | <DL | <DL | <DL | 23 ± 1 | <DL |
| | STR | 6.20 ± 0.06 b | 0.051 ± 0.001 ab | 0.062 ± 0.003 ab | 0.77 ± 0.06 | 58 ± 2 c | 409 ± 27 | 84 ± 1 b | 10 ± 1 | 98 ± 6 a | 13 ± 1 | <DL | <DL | <DL | 22 ± 1 | <DL |
| | NPK | 6.24 ± 0.01 b | 0.061 ± 0.001 b | 0.061 ± 0.001 b | 0.73 ± 0.04 | 51 ± 4 b | 381 ± 25 | 54 ± 2 a | 12 ± 4 | 139 ± 4 b | 11 ± 0 | <DL | <DL | <DL | 21 ± 1 | <DL |
| | MAP | 6.05 ± 0.02 a | 0.055 ± 0.001 ab | 0.053 ± 0.001 ab | 0.67 ± 0.06 | 53 ± 2 bc | 419 ± 18 | 58 ± 1 a | 10 ± 0 | 101 ± 3 a | 12 ± 1 | <DL | <DL | <DL | 21 ± 1 | <DL |
| | SSP | 5.94 ± 0.05 a | 0.082 ± 0.011 c | 0.052 ± 0.001 a | 0.64 ± 0.06 | 58 ± 1 c | 484 ± 16 | 58 ± 1 a | 10 ± 1 | 100 ± 3 a | 13 ± 0 | <DL | <DL | <DL | 22 ± 0 | <DL |

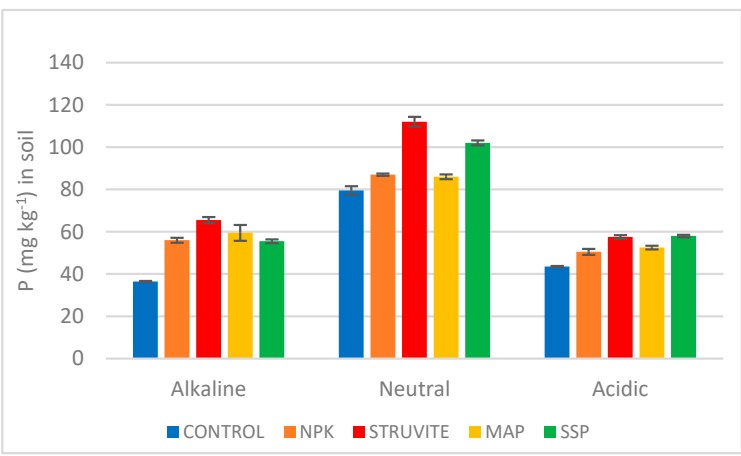

**Figure 3.** Available P content (mg/kg) in soils after leaching assay.

In the neutral soil, fertilizer application led to a significant increase in pH and N levels in all cases. A significant increase in P content was observed in the SSP and STR treatments, being more marked in the latter. No significant differences were observed in relation to the other nutrients and heavy metals.

In the acidic soil, the STR and NPK treatments significantly increased soil pH. Moreover, NPK and SSP significantly increased soil EC. The application of NPK also significantly increased N and K content. All treatments led to statistically significant increases in soil P content, the largest increase being provided by STR. This fertilizer also significantly increased the Mg content of the acidic soil (Table 5). As in the alkaline and neutral soil, no differences in Ca, Na, K, or heavy metals were observed between the different treatments.

## 4. Discussion

Agriculture is a major contributor to the eutrophication of surface water and the pollution of groundwater, mainly as a result of the overuse of fertilizers [44,45]. N is the main polluting nutrient for water, but an excess of P in waterways and coastal zones also leads to eutrophication, and this mineral is therefore regarded as a pollutant. In this context, a detailed understanding of the behavior of fertilizers in the soil to which they are applied is required.

To evaluate phosphate leaching when P fertilizers are used under conditions of high water supply to the soil, column leaching assays allow information to be obtained after a continuous supply of water forcing the movement of soluble salts. Given that soil properties condition the movement of the water after irrigation, we used three soils with distinct characteristics in this assay.

The phosphorous fertilizers evaluated differed significantly in behavior depending on the type of soil. In the alkaline soil, all P fertilizers showed very similar behavior, leaching a low amount of P compared to non-fertilized soil, corresponding with the highest concentration to the NPK treatment but without significant differences. $PO_4^{3-}$ concentrations in the leachates were much lower than those obtained from neutral or acidic soils, thereby reinforcing that P fertilizers show low solubility in calcareous soil. Two of the main factors influencing the level of available P in soil are pH and the concentration of Ca, which can precipitate with P ions [46,47]. In alkaline soil, free calcium carbonate ($CaCO_3$) buffers the soils with a pH between 7.5 and 8.5 [48] and inhibits the dissolution of primary P minerals, i.e., apatite [49]. In addition, P fixation in calcareous soils is a frequent process mainly attributed to the formation of insoluble calcium phosphates by the reaction of P with carbonates and other calcium salts. These actions overlap with other ones, such as reactions with clays, adsorption, and ionic changes [50].

In neutral soil subjected to a high water supply, differences were observed in the behavior of the distinct fertilizers. NPK-treated soil leached more than twice the amount of $PO_4^{3-}$ than STR-treated soil. The MAP and SSP treatments leached 58 and 39% more

$PO_4^{3-}$ than STR, which leached a similar amount of phosphate as the control soil. In neutral soil, the main factor influencing the leaching of phosphate ions was the solubility of the fertilizers in water, which was much lower for STR than in the other P fertilizers (Table 2), in accordance with results obtained by other authors [42].

In acidic soil, the differences were even more pronounced. The fertilizer that produced the least leaching of phosphate ions was again STR, which showed very similar behavior to the control soil. The NPK and MAP treatments leached about three times more $PO_4^{3-}$ than STR. SSP leached less phosphate, exceeding phosphate leaching from STR by about 30%. The amount of leached P ions was about three times more in acidic soil than in neutral soil due to the higher solubility of the fertilizers at low pH.

In all three soil types, significantly higher leaching of chloride ($Cl^-$) and sulfate ($SO_4^{2-}$) ions was observed when NPK was used compared to the control. Thus, for alkaline, neutral, and acidic soils, the leaching of $Cl^-$ was 3.9, 3.9, and 3.2 times higher than the control soil, respectively, and the leaching of $SO_4^{2-}$ was 12.4, 9.2, and 2.1 times higher than the control soil. $Cl^-$ and $SO_4^{2-}$ leaching was also significantly higher for SSP than for STR and the control, although to a lesser extent than NPK (1.3 and 1.12 times higher for neutral and acidic soil than control soil, respectively, for $Cl^-$; 5.7, 6 and 1.32 times higher for alkaline, neutral, and acidic soil than control soil, respectively, for $SO_4^{2-}$). Moreover, in NPK-fertilized acidic soils, a slightly higher K leaching was detected, probably due to the composition of the fertilizer.

This higher leaching of soluble salts, mainly in the case of NPK, was reflected by a significantly higher EC in the first leachates compared to that obtained in control soil and with the use of the other P fertilizers in all three types of soil. Other authors have also reported the high EC in the first phases of the leaching process when complex fertilizers, such as NPK, are used, even when these are in the controlled release category [51,52]. This observation is attributed to the presence of highly soluble salts in its composition. The reduction of salts in the leachates when STR is applied minimizes the eutrophication risk and contributes to better utilization of the nutrients supplied by the fertilizer.

The analysis of the soils after the leaching process showed differences in available P content. In all three soils, the highest available P content was found in those treated with STR, and the results were consistent with those obtained in the leaching assay. Thus, in STR-treated alkaline soil, after the leaching of soluble salts, a slightly higher available P content was observed compared to other P fertilizers.

STR and MAP showed a similar content of available P significantly higher than NPK and SSP. In neutral soil, STR-treated soil showed an available P content similar to soil fertilized with SSP and significantly higher than soil treated with NPK and MAP, which caused a higher loss of phosphate during the leaching assay. In this regard, soil from the NPK and MAP treatments showed a similar concentration of available $PO_4^{3-}$ than untreated soil. Acidic soil treated with STR showed a slight increase in available P, similar to what occurred in the SSP treatment and significantly higher than that recorded in the NPK and MAP treatments.

Of note, after the leaching process, the EC increased in the acidic and alkaline soil fertilized with NPK and SSP, and STR provided a significant supply of Mg to the acidic soil; this fact has been observed by other authors when STR is applied to the soil [27].

The results of the column leachate assay indicate that the phosphate leaching capacity of the fertilizers followed the order MAP ~ NPK > SSP > STR in the three tested soils and was in accordance with their rate of water solubility. The leaching rate of phosphate salts differed depending on the soil pH. This observation can be explained by the distinct solubility of the fertilizers at different pH, increasing as pH decreases, according to Degryse et al. (2017) [42], who found that the rate of STR dissolution was low in alkaline soils and higher in acidic soils, in both cases being much lower than the dissolution rate of MAP, with which it was compared.

Our results suggest that granular STR released P more slowly than conventional P fertilizers, confirming that STR acts as a slow-release P fertilizer, as proposed by other authors [42,43].

Conventional mineral P fertilizers are widely soluble and can cause high concentrations of P in soil leachate when rain falls soon after fertilizer application, thereby increasing the risk of eutrophication in receiving water bodies. The nutrient that contributes the most to the eutrophication of water is N, and agronomically, the amount of P lost per year in runoff is generally small. However, from the perspective of water quality, only very small concentrations of P are necessary for a body of water to become eutrophic [53]. Although significant, the agricultural transfer of P to water bodies and its contribution to the eutrophication process is lower than the urban contribution through wastewater [54].

Thus, Ott and Rechberger (2012) [55] reported a net per capita consumption in the EU15 of 4.7 kg P/yr, of which only 0.77 was recycled. A significant part of the excess P accumulated in agricultural soils (2.9 kg P/cap/yr) was emitted to the hydrosphere (0.55 kg P/cap/yr). These figures reflect the need to optimize the use of P fertilizers and recycle P-rich wastes by implementing P recovery techniques in wastewater treatment plants. Such a strategy could significantly reduce Europe's dependence on imports of this mineral. In this regard, the recovery of STR from wastewater would imply a double benefit. First, it would allow a reduction in the eutrophication of water by reducing the P load in the treated urban wastewater that drains into open waters, as other authors have concluded [56]. Second, this approach would bring about the recovery of a product that can be used as a slow-release P fertilizer that is characterized by a low content of heavy metals and the capacity to reduce the leaching of nutrients and contaminants in comparison with commonly used commercial fertilizers. Moreover, the use of STR would contribute to reducing the dependence on phosphate rock by using a product from wastewater treatment in line with the objectives of the circular economy.

## 5. Conclusions

The STR application showed low leaching of phosphorus salts in the three types of soil compared to the conventional fertilizers used. Thus, the use of STR as a P fertilizer would reduce P leaching and, consequently, the risk of eutrophication derived from the high solubility of conventional fertilizers, thus making this product an environmentally safe P fertilizer.

After leaching events, STR application increases the concentration of available P in the soil in concentrations comparable to the rest of the fertilizers and higher in the case of soil with a neutral pH. Furthermore, STR supplies N and Mg to crops. However, the application of mineral fertilizers must be carried out according to the characteristics of each soil to avoid the risk of salt leaching.

The use of STR as a fertilizer implies the recovery of a high-value resource (P) from wastewater treatment plants and, thus, complies with the principles of the circular economy.

**Supplementary Materials:** The following supporting information can be downloaded at: https://www.mdpi.com/article/10.3390/environments10020022/s1, Figure S1: Amount of NO3- leached from alkaline (A), neutral (B) and acidic (C) soil columns as a function of time in struvite and P-fertilizers., Figure S2: Amount of Cl- leached from alkaline (A), neutral (B) and acidic (C) soil columns as a function of time in struvite and P-fertilizers., Figure S3: Amount of SO42- leached from alkaline (A), neutral (B) and acidic (C) soil columns as a function of time in struvite and P-fertilizers.

**Author Contributions:** Conceptualization, C.M. and M.C.L.; Methodology, J.A.; Validation, J.A.; Formal Analysis, S.D.-P. and J.A.; Investigation, C.M., M.G.-D.; Resources, M.C.L.; Data Curation, J.A., C.M.; Writing—Original Draft Preparation, C.M.; Writing—Review and Editing, C.M., M.G.-D., and M.C.L.; Project Administration and Funding Acquisition: M.C.L. All authors have read and agreed to the published version of the manuscript.

**Funding:** This research was funded by Canal de Isabel II, S.A. (project STRUVITE).

**Data Availability Statement:** Not applicable.

**Acknowledgments:** This work was supported by the research project STRUVITE (Canal de Isabel II, 2020–2022).

**Conflicts of Interest:** The authors declare no conflict of interest.

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
