# Peer review of "Assessment of Recovered Struvite as a Safe and Sustainable Phosphorous Fertilizer"

_environments, doi:10.3390/environments10020022_

Round 1

Reviewer 1 Report

Manuscript covers an interesting area of research, however, there is always room for improvement. File is attached for making correction.

Author Response

Please, see the attachment file with the response to Reviewer 1.

Reviewer 2 Report

Dear authors! The study was conducted on a topical issue of high scientific and practical importance. A large amount of research has been carried out.

However, there are certain shortcomings that reduce the quality of the work done and the reliability of the results obtained.

1. When conducting research, the granulometric composition of soils was not indicated. No effect of granulometric composition on the loss of phosphorus during leaching was revealed.

2. The type of soils studied, their typicality and distribution for the study region are not indicated.

3. Aluminum and iron compounds have a great influence on the mobility of phosphates in the soil, which was not studied in the study.

4. In alkaline soils, the content of diphosphate ions, which predominate at pH above 7.0, has not been studied.

Author Response

Please, see the attachment file with the response to Reviewer 2.
